# Lipid Raft Integrity and Cellular Cholesterol Homeostasis Are Critical for SARS-CoV-2 Entry into Cells

**DOI:** 10.3390/nu14163417

**Published:** 2022-08-19

**Authors:** Ahmed Bakillah, Fatimah Al Hejji, Abdulrahman Almasaud, Haya Al Jami, Abbas Hawwari, Ali Al Qarni, Jahangir Iqbal, Naif Khalaf Alharbi

**Affiliations:** 1King Abdullah International Medical Research Center (KAIMRC), Eastern Region, Al Ahsa 31982, Saudi Arabia; 2King Saud bin Abdulaziz University for Health Sciences (KSAU-HS), Al Ahsa 36428, Saudi Arabia; 3King Abdulaziz Hospital, Ministry of National Guard-Health Affairs (MNG-HA), Al Ahsa 36428, Saudi Arabia; 4Vaccine Development Unit, Department of Infectious Disease Research, King Abdullah International Medical Research Center, Riyadh 11481, Saudi Arabia; 5King Saud bin Abdulaziz University for Health Sciences (KSAU-HS), Riyadh 11435, Saudi Arabia

**Keywords:** SARS-CoV-2, lipid rafts, cholesterol, methyl-β-cyclodextrin, angiotensin-converting enzyme 2, protease transmembrane protease serine 2, camostat mesylate, statins

## Abstract

Lipid rafts in cell plasma membranes play a critical role in the life cycle of many viruses. However, the involvement of membrane cholesterol-rich lipid rafts in the entry of severe acute respiratory syndrome coronavirus 2 (SARS-CoV-2) into target cells is not well known. In this study, we investigated whether the presence of cholesterol-rich microdomains is required for the entry of SARS-CoV-2 into host cells. Our results show that depletion of cholesterol in the rafts by methyl-beta-cyclodextrin (MβCD) treatment impaired the expression of the cell surface receptor angiotensin-converting enzyme 2 (ACE2), resulting in a significant increase in SARS-CoV-2 entry into cells. The effects exerted by MβCD could be substantially reversed by exogenous cholesterol replenishment. In contrast, disturbance of intracellular cholesterol homeostasis by statins or siRNA knockdown of key genes involved in the cholesterol biosynthesis and transport pathways reduced SARS-CoV-2 entry into cells. Our study also reveals that SREBP2-mediated cholesterol biosynthesis is involved in the process of SARS-CoV-2 entry in target cells. These results suggest that the host membrane cholesterol-enriched lipid rafts and cellular cholesterol homeostasis are essential for SARS-CoV-2 entry into cells. Pharmacological manipulation of intracellular cholesterol might provide new therapeutic strategies to alleviate SARS-CoV-2 entry into cells.

## 1. Introduction

Virus entry into target cells is a complex process that occurs via mechanisms involving both cellular and viral components, with the main goal to escape the host defense. Many viruses such as severe acute respiratory syndrome coronavirus 2 (SARS-CoV-2) achieve this process by acquiring a lipid envelope during exit from host cells which allows them to enter target cells by fusion of their envelope with the host plasma membrane at the cell surface or alternatively take the advantage of the endocytic pathway [1,2]. Angiotensin-converting enzyme 2 (ACE2) has been reported as the cellular receptor for the enveloped coronaviruses SARS-CoV and SARS-CoV-2 [3,4]. SARS-CoV-2’s preferential entry into target cells depends on the target-cell proteases cathepsin L and protease transmembrane protease serine 2 (TMPRSS2) that are involved in S-protein activation [5]. TMPRSS2 is present at the cell surface, and TMPRSS2-mediated S-protein activation occurs at the plasma membrane, whereas cathepsin L-mediated activation occurs in the endo-lysosomes [6].

Lipid rafts are specialized membrane microdomains that are enriched in cholesterol, sphingolipids, and associated proteins such protein kinases and glycosylphosphatidylinositol (GPI)-anchored proteins. Lipid rafts are a dynamic platform associated with the regulation of various physiological and biochemical processes such as membrane protein trafficking, signal transduction pathways, neurotransmission, cell survival and proliferation, apoptosis, cell adhesion, and membrane fluidity [7,8,9,10]. Many viruses and pathogens require cholesterol-rich lipid rafts to enter and exit target cells though the plasma membrane [11,12,13,14,15,16].

The cholesterol of cell membranes is tightly regulated, and its content controls cellular lipid rafts’ stability. Any change in the cholesterol content modifies their physical and biological properties. In fact, many studies have shown that lipid rafts are disrupted by cholesterol depletion, leading to loss of membrane integrity and delocalization of raft components [17]. Cholesterol has been recently recognized to be involved in SARS-CoV-2 entry into target cells [18,19,20,21]; however, plasma membrane cholesterol-rich raft formation and rearrangement in the host cells have not been fully explored.

Accumulating evidence suggests that enveloped virus entry may require cholesterol in either the viral envelope or plasma membrane rafts, or in both, for efficient membrane fusion. For example, human immunodeficiency virus (HIV) requires cholesterol in both the target cell membrane and the viral envelope [11,22], while for influenza virus and duck hepatitis B virus, the presence of cholesterol in the viral envelope is critical, but not essential, in the host’s plasma membrane [23,24]. In contrast, some other viruses such vesicular stomatitis virus, dengue virus, and yellow fever virus enter target cells independently of cholesterol [25].

Although there is abundant information on the cell entry of SARS-CoV-2, there is still a lack of detailed understanding of the fusion spike protein–cholesterol interaction mechanism, and how the virus utilizes plasma cholesterol-enriched rafts to enter the host target after binding to the ACE2 receptor. Here, we addressed the role of cellular cholesterol during the early stage of SARS-CoV-2 entry in target cells.

We used independent approaches to manipulate cholesterol biosynthesis to investigate underlying mechanisms of SARS-CoV-2 entry into cells. We used cholesterol-chelating agents to deplete the cholesterol-enriched plasma membrane of the host target. We applied lipid-lowering drugs that pharmacologically interfere with the relevant enzymes involved in cholesterol biosynthesis and transport. Finally, we used siRNA-mediated silencing target genes of cholesterol synthesis. Both genetic manipulation and pharmacological inhibition of cholesterol synthesis caused a decrease in SARS-CoV-2 entry in host cells. Our results demonstrate, for the first time, that plasma membrane lipid raft integrity and intracellular cholesterol homeostasis are critical for SARS-CoV-2 entry into cells.

## 2. Materials and Methods

### 2.1. Cell Lines and Reagents

Vero-E6, HepG2, Huh7, Caco2, THP1, and A549 cell lines were obtained from the Core Laboratories Facility of King Abdullah International Medical Research Center (KAIMRC) in Riyadh, Saudi Arabia. Except for THP1 cells, which were grown in RPMI-1640 medium, all the other cell lines were maintained in Dulbecco’s modified Eagle’s medium (DMEM) containing low glucose, 10% fetal bovine serum, L-glutamine, and 1% penicillin/streptomycin. For experiments, cells from 70 to 80% confluent flasks (75 cm^2^ flasks, Corning Glassworks, Corning, NY, USA) were seeded into 6-well, 12-well, or 24-well plates. Methyl-β-Cyclodextrin (MβCD; catalog #128446-36-6), nystatin (catalog #N6261), and cholesterol (catalog #57-88-5) were purchased from Sigma-Aldrich (Sigma-Aldrich, St. Louis, MO, USA). Myriocin (ISP-I; catalog #ab143526), aloxistatin (E64d, catalog #144048), camostat mesylate (catalog #ab145709), and T0901317 (catalog #142808) were purchased from Abcam (Abcam plc, Cambridge, UK). Statins (atorvastatin, catalog #CAS134523-00-5; simvastatin, catalog #CAS79902-63-9; and pravastatin, catalog #CAS81131-70-6) were purchased from Santa Cruz Biotechnology Sigma Aldrich (St. Louis, MO, USA). ACE2 and TMPRSS2 antibodies were purchased from Novus Biologicals (Novus Biologicals, LLC, Cambridge, UK). siRNAs were purchased from Dharmacon (Pittsburg, PA, USA). TRIzol^TM^, catalog #15596018 was purchased from Life Technologies (Carlsbad, CA, USA). A High-Capacity cDNA reverse transcription kit (catalog #4368813) was purchased from ThermoFisher Scientific (Waltham, MA, USA). A qPCR^TM^ core kit for SYBR Green I,catalog #10-SN10-05 was purchased from Eurogentec (San Diego, CA, USA). Primary and secondary antibodies were purchased from either Cell Signaling (Danvers, MA, USA) or Abcam (Cambridge, MA, USA). All other chemicals and solvents were obtained from Fisher Scientific through its local distributor in the Kingdom of Saudi Arabia.

### 2.2. SARS-CoV-2pp Entry and Cell Viability Assays

SARS-CoV-2 pseudotyped particles (SARS-CoV-2pp) were produced in HEK293T cells and titrated using Huh7.5 cells as described previously [26,27]. For virus entry evaluation, cells were seeded into 24-well plates and infected with varying concentrations of SARS-CoV-2 spike-bearing pseudoviruses in medium containing 10% serum for 24–48 h at 37 °C. After 24–48 h post-infection, the supernatant was removed, and cells were washed twice with PBS. Cells were then lysed with 30 µL lysis buffer followed by addition of 50 µL of Bright Glo^TM^ luciferase substrate (catalog #E2610; Promega, Madison, WI) at 1:1 dilution in prewarmed complete cell growth media. After 5 min of incubation at room temperature, the supernatant was transferred into a 96-well opaque plate. The relative luminescence units (RLUs) of Luc activity were measured using a luminescence microplate reader (Agilent BioTek Synergy H1, Winooski, VT, USA). To determine the cell viability following infection and treatment, cells were incubated with Cell Titer-Glo^®^ assay reagent according to the manufacturer’s instructions (Promega, Madison, WI). The relative luminescence units (RLUs) corresponding to the amount of ATP present in metabolically active cells were measured using a multimode microplate reader (GloMax^®^ Explorer System, Promega, Hampshire, UK).

### 2.3. Cell Treatment with Cholesterol-Distrupting Agents and Statins

Cells were treated with varying concentrations of MβCD (2–10 mM) for 1 h at 37 °C or nystatin (1–20 µM) for 24 h at 37 °C diluted in serum-free media. After incubation, cells were washed once with fresh media prior to infection with SARS-CoV-2pp as described above. For cholesterol replenishment, cells were pretreated with 10 mM MβCD for 1 h at 37 °C, and after washing, cells were incubated with exogenous cholesterol alone (0.25 mM) or MβCD–cholesterol complexes (10 mM:0.125 mM) [28] in serum-free media for 1 h at 37 °C. After washing, cells were infected with pseudoviruses for 48 h at 37 °C in media containing 10% serum. For the statin treatment, cells were incubated with varying concentrations of different statins (0.5–10 µM) for 48 h at 37 °C in media containing 10% serum prior to the infection procedure.

### 2.4. Preparation of Wild-Type SARS-CoV-2

SARS-CoV-2 was isolated from a confirmed COVID-19 patient’s nasopharyngeal swab, in early 2020, which can be considered as one of the ancestral SARS-CoV-2 strains and not any of the subsequent variants. The patient sample was processed in a BSL-3 facility, vortexed, and spun, and the supernatant was filtered through a 0.22 µM filter. Then, 200 µL of filtered samples was added to a monolayer of Vero cells in a 25 cm flask. Three days post-inoculation, the cytopathic effect (CPE) was observed under a light microscope, and 200 µL of supernatant was passaged in a 75 cm flask of Vero cells for the second and third passages. The virus was then amplified in 175 cm flasks (passage 4) and purified by regular centrifugation and filtration in a total volume of 180 mL. The purified virus from each passage was confirmed by RT-PCR targeting the three genes of SARS-CoV-2 using the primers and probes listed in Appendix A.

The virus titer was determined, and the virus stock was frozen at −80 °C in small aliquots until further use.

### 2.5. Cholesterol Depletion and Replenishment Using Wild-Type SARS-CoV-2

Huh7 and Vero E6 cells were pretreated with MβCD (2 mM and 5 mM) or cholesterol (0.25 mM) in serum-free DMEM for 1 h. After treatment, cells were washed once with fresh media and incubated with SARS-CoV-2 at MOI = 0.1 for 1 h at 37 °C. Media containing the remaining inoculum were removed and replaced with complete growth DMEM with 10% FBS. Media (200 µL) from each well were collected at 24 h and 48 h post-infection (p.i.) and used for RNA isolation and qRT-PCR analysis for viral titer measurement, as described previously [29,30].

### 2.6. Intracellular Cholesterol Quantification

Huh7 and Vero E6 cells grown in 24-well plates were infected as described above. Intracellular cholesterol was determined as described elsewhere [31]. Briefly, at 48h p.i., after washing the cells 3 times with cold PBS, 1 mL of hexane/isopropanol (2:1, *v**/v*) was added to each well for 30 min of incubation at room temperature. The organic solvent was transferred to tubes and placed under a hood for evaporation under nitrogen. The extracted lipids were resuspended in 100 µL of isopropanol containing 10% Triton X-100. The content of cellular cholesterol was measured using commercially available kits from Thermo Scientific (Middletown, VA, USA), as described previously [32]. The remaining cells in the wells after lipid extraction were resuspended in 0.1N NaOH, and the protein content was measured using the Pierce™ Coomassie (Bradford, UK) protein assay kit from Thermo Scientific (Middletown, VA, USA).

### 2.7. Western Blot Analysis

For detection of proteins, cells were lysed with RIPA buffer and separated on 4–20% Mini-PROTEAN TGX precast protein gels (catalog #4561096) from BioRad (Hercules, CA, USA). Separated proteins were transferred to a PVDF membrane, blocked with 50 mM Tris, pH 7.6, 150 mM NaCl, 0.5% Tween-20, and 5% milk (TBS plus Tween-20), and probed with anti-human ACE2 and TMPRSS2 antibodies (1:1000 dilution) overnight at 4 °C, followed by incubation with horseradish peroxidase-conjugated goat anti-mouse or anti-rabbit IgGs (1:5000 dilution) for 1 h at room temperature. The blots were developed with the Clarity Western ECL substrate (catalog #1705060) from BioRad (BioRad, Hercules, CA, USA). The results were photographed with the ChemiDoc MP Imaging System (BioRad, Hercules, CA, USA).

### 2.8. Immunofluorescence Staining Analysis

Vero E6 and Huh7 cells were cultured on coverslips in 6-well plates and treated with MβCD and cholesterol as described above. At 48 h p.i. with the wild-type SARS-CoV-2, cells were washed with PBS and fixed with 4% formalin (15–20 min), permeabilized with 0.3% triton X-100 (10 min) and blocked with 3% BSA. After washing with PBS, cells were probed with primary antibodies overnight at 4 °C (anti-spike protein monoclonal antibody; Cell signaling, catalog #99423; 1:500 dilution). After three washes with PBS, spike proteins were detected with Alexa Fluor 488 goat anti-mouse antibodies (Abcam, catalog #ab150113). For F-actin detection, cells were incubated with fluorescent phalloidin solution for 30 min (Abcam catalog #ab235137). Coverslips containing the cells were mounted on glass slides using aqueous mounting media. Cells were observed using phase-contrast and fluorescence (3D Cell Explorer, NanoLive SA, Tolochenaz, Switzerland) microscopy, and photographs were taken.

### 2.9. RNA Isolation and Quantitative Real-Time PCR (qRT-PCR)

Total RNA from cells was isolated using TRIzol^TM^ (catalog #15596018; Life Technologies (Carlsbad, CA, USA) as per the manufacturer’s instructions. The purity of RNA was assessed by the *A**_260_/A_280_* ratio. RNA preparations with *A**_260_/A_280_* ratios of more than 1.7 were used for cDNA synthesis. The first-strand cDNA was synthesized using a High-Capacity cDNA reverse transcription kit. Each reaction of quantitative PCR was carried out in a volume of 20 µL, consisting of 5 µL of cDNA sample (1:25 dilution of the first-strand cDNA sample) and 15 µL of SYBR Green I PCR master mix solution containing 1X PCR buffer. The PCR was carried out by incubating the reaction mixture first for 10 min at 95 °C followed by 40 cycles of 15 s incubations at 95 °C and 1 min at 60 °C in a QuantStudio™ 6 Flex Real-Time PCR (Applied Biosystems). Data were analyzed using the ΔΔC_T_ method, according to the manufacturer’s instructions, and presented as arbitrary units that were normalized to the expression of actin or GAPDH. The list of the sequences of primers used in this study is shown in Appendix A.

### 2.10. SiRNA Transfection

On-TARGET plus Smartpool human siRNAs against the *SREBP2* and *ABCA1* genes were obtained from Dharmacon (Pittsburgh, PA, USA). Huh-7 cells were transfected with siRNA at a final concentration of 100 nM using the DharmaFECT1 transfection reagent. Non-targeting scrambled siRNAs were used as controls and were denoted as NT-siRNA. Following the 48 h transfection, cells were infected with SARS-CoV-2pp and collected 48 h p.i. for RNA isolation, qRT-PCR, and luciferase activity measurement.

### 2.11. Statistical Analysis

Data are presented as the mean ± SD from at least three independent experiments using GraphPad Prism Software (version 5.0; GraphPad, San Di-ego, CA, USA). The statistical analysis of differences between groups was performed using a two-tailed Student *t*-test with a confidence level of 95% or two-way analysis of variance (ANOVA) for multiple comparisons where appropriate. A *p* value of < 0.05 was considered significant.

## 3. Results

### 3.1. SARS-CoV-2pp Entry into Cells Is Dependent on ACE2 Expression Levels

To initiate our studies, we examined the entry of SARS-CoV-2pp into cells, where several cell lines were infected with three independent batches of virus preparations (yield ranging from 1 × 10^6^ to 10 × 10^6^ RLU/mL) at different doses of the virus and times post-infection. Among the tested cell lines, Vero E6 cells were the most responsive cells to pseudoviral infection after 48 h p.i., most likely due to their higher expression of the *ACE2* receptor (Figure 1A,B). Despite the low expression of *ACE2* in HepG2 and A549 cells, SARS-CoV-2pp was able to enter these cells with a similar magnitude to Huh-7 and CaCo2 cells (Figure 1B). These data show that the efficiency of SARS-CoV-2pp entry into cells depends on the level of cellular expression of ACE2 receptors.

Subsequently, Vero E6 (high-*ACE2*-expressing cells) and Huh7 (low-*ACE2*-expressing cells) cells were infected with increasing doses of SARS-CoV-2pp for 48 h. The entry of the virus increased in a dose-dependent manner, with the concentration of the virus reaching its maximum entry with a dose of 300,000–400,000 RLU/mL (Figure 2AB), without any significant effect on cell viability, as measured by the ATP levels (Figure 2C,D).

### 3.2. Protease Inhibitors Attenuated SARS-CoV-2pp Entry into Cells

The serine protease inhibitor camostat mesylate prevents viral fusion activation by TMPRSS2, whereas E64d inhibits cathepsin B/-mediated fusion after virus uptake by endocytosis [33]. To further validate the SARS-CoV-2pp entry assay, the inhibitory activities of these two clinically proven inhibitors were evaluated. Treatment of Huh7 cells with camostat and E64d (1–20 µM) significantly inhibited SARS-CoV-2pp entry by 50–70% and 52–87%, respectively (Figure 3A), without any apparent cellular cytotoxic effects (Figure 3B). These data demonstrate that this optimized assay could be used as an effective and convenient assay to screen potential inhibitors for SARS-CoV-2pp entry.

### 3.3. Depletion of Cholesterol Alters the Expression of Raft-Associated Receptors That Are Involved in Cell Entry of SARS-CoV-2pp

MβCD is known to deplete cholesterol from plasma membranes, thereby disrupting lipid raft components [17]. We, therefore, examined the effect of MβCD-mediated cholesterol depletion from the cell plasma membrane on raft-associated receptors. HepG2, Huh7, and Vero E6 cells were treated with MβCD (10 mM), cholesterol alone (0.25 mM), or MβCD–cholesterol (10 mM:0.125 mM) for 1h followed by SARS-CoV-2pp infection for 48 h. None of these treatments affected ACE2 expression, measured by the cellular mRNA levels of *ACE2*, as compared to untreated cells (Figure 4A). MβCD treatment significantly increased SARS-CoV-2pp entry into Huh7 cells (278%) and Vero E6 cells (320%) (Figure 4B). Supplementation of cholesterol attenuated the cell entry of SARS-CoV-2pp into Huh7 and Vero E6 cells (Figure 4B). In contrast, MβCD treatment dramatically reduced the expression of the ACE2 protein in both Huh7 and Vero E6 (Figure 4C). Replenishment of cells by exogenous cholesterol partially reversed the effect of MβCD in Huh7 and Vero E6 cells (Figure 4C). In subsequent experiments, MβCD treatment resulted in a significant decrease in intracellular cholesterol in the cells (Appendix A). The reduction in intracellular cholesterol reached its maximum (~50%) at 2–60 min of treatment (Appendix A, Panel A). At 10 mM MβCD, intracellular cholesterol was reduced by 40–60% and 40–80% in Huh7 and Vero cells, respectively. Replenishment by MβCD–cholesterol complexes reversed the MβCD-induced cholesterol reduction (Appendix A, Panels B,C). These data suggest that membrane integrity and the presence of functional receptors on cholesterol-rich rafts are crucial for productive entry of SARS-CoV-2pp into the cells.

### 3.4. Nystatin Treatment Enhances Cell Entry of SARS-CoV-2pp

To confirm the role of plasma cholesterol-rich rafts in SARS-CoV-2pp entry, Huh7 cells were treated with another sequestering agent that binds to cholesterol, preventing the formation of rafts, namely, nystatin (1 µM–20 µM), for 48 h prior to pseudoviral infection. Nystatin treatment increased the entry of SARS-CoV-2pp in Huh7 cells by 215% and 482% at 10 µM and 20 µM, respectively (Figure 5A), without affecting the cell viability (Figure 5B). These results indicate that functionally intact cholesterol-rich rafts are necessary in preventing the cell entry of SARS-CoV-2. Any alteration in the integrity of plasma membrane lipid rafts may lead to increased entry of SARS-CoV-2 into cells.

### 3.5. Depletion of Cholesterol-Rich Rafts Enhanced Cell Entry of the Wild-Type SARS-CoV-2 without Altering Cell Morphology

To further validate our findings with the pseudotyped viruses, MβCD-treated cells were infected with wild-type SARS-CoV-2 (MOI = 0.1). MβCD treatment significantly increased the viral load in Huh7 and Vero E6 cells, and this effect was attenuated by cholesterol replenishment in Huh7 cells (Figure 6A–C), but not in Vero E6 cells (Figure 6B–D). Although the immunofluorescence is a semi-quantitative assay, MβCD treatment (2 mM) enhanced the entry of wild-type SARS-CoV-2 in Huh-7 and Vero E6 cells without any apparent morphological and structural changes in cells (Appendix A). Taken together, these data confirm that, similar to SARS-CoV-2pp, MβCD enhanced wild-type virus entry into the cells, most likely due to alterations in membrane integrity and delocalization of cellular receptors from raft domains.

### 3.6. Impairment of Intracellular Cholesterol Homeostasis Mitigates SARS-CoV-2pp Infection

One possible mechanism to increase cholesterol-rich raft formation is increasing the synthesis of the cellular cholesterol content. Statins are known for lowering cholesterol synthesis by inhibiting the activity of HMG-CoA reductase, which is the rate-limiting enzyme in the cholesterol/mevalonate pathway. The liver X receptors (LXRs) are responsible for the transcriptional regulation of a number of genes involved in cholesterol efflux from cells and therefore may impact the supply of cholesterol to lipid rafts. To investigate whether statins and LXR agonists could modulate SARS-CoV-2pp entry, cells were treated with various classes of statins and the LXR agonist T0901317 for 48 h prior to cell infection with SARS-CoV-2pp, which was followed for another 48 h. The most commonly prescribed statins, atorvastatin and simvastatin, significantly reduced SARS-CoV-2 entry at 10 µM by ~56% (Figure 7A). The effect of pravastatin was more pronounced and decreased the virus entry at all doses (0.6–10 µM) by ~60% (Figure 7A). Similarly, the LXR agonist T0901317, a non-steroidal synthetic potent compound that promotes cholesterol efflux from the cells, significantly reduced the virus entry by ~50% (Figure 7A). There were no changes in cell viability, indicating the absence of major cytotoxic effects of statin treatment (Figure 7B). Under these conditions, statins decreased the secretion of apolipoprotein B-containing lipoproteins (Figure 7C) and the intracellular cholesterol content (Figure 7D) by 50–65% and 50–80%, respectively. Collectively, our data highlight that disruption of intracellular cholesterol levels by statins and LXR agonists attenuates SARS-CoV-2pp entry into cells.

### 3.7. SARS-CoV-2pp Entry into Cells Alters Cholesterol- and Lipid-Modifying Gene Expression

To determine if our findings were the result of changes in the transcriptional regulation of genes involved in cholesterol and lipid pathways, transcript levels of selected relevant genes were quantified using qRT-PCR in SARS-CoV-2-infected Huh7 cells. SARS-CoV-2pp entry into Huh7 cells upregulated the gene expression of cholesterol metabolic pathways such as *SREBP2*, *HMG-CoA synthase*, and *HMG-CoA reductase* (Figure 8). Additionally, cholesterol-25-hydroxylase (*CH25H*), which codes the enzyme that synthesizes oxysterol-25hydroxysterol (25HC), was significantly downregulated in SARS-CoV-2-infected cells. This finding suggests that SREBP2-mediated cholesterol biosynthesis may play a role in SARS-CoV-2pp infection of cells.

### 3.8. siRNA-Mediated Silencing of SREBP2 and ABCA1 Genes Reduced SARS-CoV-2pp Entry into Cells

To shed more light on the modulation of target genes involved in the regulation of cholesterol homeostasis, the expression of SREBP2 (sterol sensor) and ABCA1 (cholesterol transporter) was knocked down using siRNA-mediated silencing. Transfection of Huh7 cells with siRNA targeting the SREBP2 and ABCA1 genes significantly reduced transcript levels of SREBP2 and ABCA1, as assessed by qRT-PCR (Figure 9A). This was associated with significant changes in the mRNA levels of the downstream target genes HMG-CoA reductase and LDLR (Figure 9B,C). Silencing the SREBP2 and ABCA1 genes significantly reduced SARS-CoV-2pp infection in Huh7 cells by 43% and 62%, respectively (Figure 9D). Taken together, these data demonstrate that the SREBP2 and ABCA1 genes play an important role in cholesterol biosynthesis and transport, which are required by SARS-CoV-2pp to enter cells.

## 4. Discussion

Cholesterol is a major component of lipid raft microdomains. Lipid rafts are traditionally described as a specialized dynamic platform within plasma membranes that are enriched in cholesterol, sphingolipids, and receptors that are involved in various biological processes [9,34,35]. The role of cholesterol has attracted increasing attention for understanding the process of SARS-CoV-2 infection [21,36,37,38]. Cholesterol is essential for the assembly, replication, and infectivity of enveloped virus particles [19]. It has been reported that the increased mortality of severe and critical COVID-19 patients was significantly associated with a reduction in circulating LDL cholesterol [39]. Another observational study showed that patients with severe COVID-19 had lower HDL cholesterol and higher triglyceride levels [40]. Additionally, a study reported that the use of statins during the 30 days prior to admission for COVID-19 was associated with a low risk of developing complications and a faster time to recovery for patients without severe disease [41].

Many viruses and pathogens utilize plasma rafts in their life cycle to enter, replicate, and exit the host cells [42,43]. Despite recent advances in understanding SARS-CoV-2 entry into cells, the exact entry mechanism has not been fully investigated. Two distinct pathways have been proposed for SARS-CoV-2 entry using the endosomal entry route or cell surface entry route involving the ACE2 receptor and target-cell proteases TMPRSS2 and cathepsin L that are critical for the cleavage of the virus spike (S) protein at the S1/S2 subunits and the S2′ site, allowing fusion of viral and cellular membranes [6].

MβCD, a cholesterol-chelating agent, has been widely used to deplete cholesterol-rich rafts to study the infection process of many viruses. MβCD acts strictly on the cell membrane and rapidly extracts cholesterol from lipid rafts [44]. In this study, we demonstrated that MβCD-mediated cholesterol depletion of cellular rafts enhanced SARS-CoV-2 entry into cells using pseudotyped and wild-type viruses. This effect was associated with the disruption of cellular ACE2 receptors and TMPRSS2 that may have resulted in alterations in the structural integrity of the plasma membrane rafts and/or redistribution of ACE2 receptors from the point of virus entry to non-raft domains. Replenishment of cells with exogenous cholesterol substantially reversed MβCD-mediated raft changes. While there are many examples of studies reporting increased viral infectivity with MβCD-mediated depletion of cholesterol [45,46,47,48], this study is in contrast to other studies reporting a reduction in virus entry following MβCD acute treatment [18,20,49,50,51]. Another study reported that inhibition of SARS-CoV-2 infection by the MβCD-mediated depletion of cholesterol particularly affected the virus membrane but not the host plasma membrane [18]. One plausible explanation for this discrepancy could be related to differences in the cell lines used, the MβCD dose, and the treatment duration in these studies. Intriguingly, our experiment with the wild-type SARS-CoV-2 clearly revealed a differential outcome between Huh7 and Vero E6 cells for the use of exogenous cholesterol to reverse the exacerbated effect of MβCD, a difference that could be explained by the higher capacity of hepatocytes to package cholesterol and secrete it as apoB-containing lipoproteins. Studies have shown that MβCD suppresses viral replication by perturbing the accumulation of virus particles and membrane lipid rafts [52,53,54]. Hence, it is unlikely that the enhanced infectivity of wild-type SARS-CoV-2 in host cells is due to direct effect of MβCD on the replication of the virus.

Higher doses of nystatin resulted in a similar effect to MβCD, enhancing SARS-CoV-2pp entry into cells. Cholesterol is critical for membrane integrity and permeability [55,56]; therefore, any minor changes in cholesterol could result in dysfunctional raft-associated proteins. The cholesterol-sequestering agents MβCD and nystatin act on lipid rafts via different mechanisms. While MβCD sequesters the membrane cholesterol and removes it from cells [45], nystatin only binds to cellular cholesterol, forming lipid complexes and preventing the formation of lipid rafts [57]. It is also possible that the binding of nystatin to cholesterol contributes to conformational changes in ACE2 receptors, resulting in more efficient entry of SARS-CoV-2 into cells. Nevertheless, it should be noted that MβCD not only extracts cholesterol from the plasma membrane, but also from intracellular compartments, thereby altering organelle function and vesicle transport [58]. Thus, caution should be taken when interpreting studies using MβCD-mediated depletion of cholesterol, especially at a relatively high dose. The observed reduction in the cellular expression of ACE2 receptors following MβCD-mediated cholesterol depletion might not be unique to SARS-CoV-2. In fact, a study has shown that MβCD-mediated removal of cholesterol in HIV-infected macrophages resulted in a reduction in the cell surface expression of the CD4/CXCR4 and CCR5 receptors [59]. Another study showed that MβCD treatment also reduced the ACE2 receptor in SARS-CoV-infected Vero cells, as demonstrated by flow cytometry [13].

Statins are effective lipid-lowering drugs that demonstrate favorable anti-inflammatory profiles and have been recently proposed as a potential therapy for COVID-19, but clinical outcome results are conflicting [60,61,62,63,64,65,66]. In this study, we presented evidence for the critical role of cellular cholesterol homeostasis in the infection of SARS-CoV-2 in some cell lines. In addition, we identified the transcription factor SREBP2 as a key element in this process. This is consistent with a recent study reporting high activation of SREBP2 in peripheral blood mononuclear cells of COVID-19 patients, leading to cytokine storm [67]. One of the mechanisms used by cells to increase the intracellular cholesterol content is to increase the activity of key enzymes involved in the cholesterol biosynthesis pathway. Our study revealed that the use of various classes of statins that are known to be competitive inhibitors of HMG-CoA reductase, the rate-limiting enzyme for cholesterol synthesis, reduced SARS-CoV-2 entry into and infection of the cell lines. In addition, knockdown of the *SREBP2* gene, which modulates downstream target genes such as *HMG-CoA reductase*, blocked the entry of SARS-CoV-2, suggesting that intracellular cholesterol homeostasis in the host could play a critical role in SARS-CoV-2 entry.

An increase in cholesterol or 25-hydroxycholesterol (25-OH cholesterol) levels was shown to suppress the synthesis of HMG-CoA reductase and result in a remarkable decrease in HMG-CoA reductase [68]. In this study, gene expression analysis of SARS-CoV-2-infected cells revealed a marked decrease in cholesterol 25-hydroxylase (CH25H), the enzyme responsible for converting cholesterol into 25-OH cholesterol (a natural oxysterol). This finding is consistent with a recent study demonstrating that induction of CH25H was able to inhibit SARS-CoV-2 by depleting membrane cholesterol [37]. In addition to SREBP2, we also found that the entry of SARS-CoV-2 in cells induced upregulations of major genes involved in cholesterol biosynthesis including *HMG-CoA reductase* and *HMG-CoA synthase*. Surprisingly, we only found a trend increase in *ABCA1*, but it was not significant. Interestingly, knockdown of the *SREBP2* and *ABCA1* genes resulted in significant upregulation of *LDLR.* The transcription of *LDLR* is primarily under the control of SREBP2, an endoplasmic reticulum transmembrane protein that exists in a complex with the sterol-sensing protein SCAP [69]. The complex SREBP2/SCAP moves to the Golgi apparatus when the intracellular cholesterol concentration is low. The specific proteases SP1 and SP2 sequentially cleave SREBP2 to release the active N-terminal SREBP2 transcription domain, which translocate to the nucleus and activates genes involved in cholesterol synthesis and uptake such as *HMG-CoA reductase* and *LDLR* [70,71]. It is important to note that intracellular cholesterol concentrations are tightly controlled by regulatory mechanisms that keep levels of unesterified cellular cholesterol constant despite significant fluctuations in cholesterol requirements [72].

Another important gene involved in the cholesterol metabolism is *ABCA1*, a cholesterol efflux gene. Interestingly, we found that knockdown of *ABCA1* resulted in dramatic upregulation of *LDLR*, most likely to compensate for the intracellular dysregulation of cholesterol metabolism in the absence of ABCA1. The marked increase in expression levels of LDLR in ABCA1-silencing SARS-CoV-2-infected cells is consistent with a study using hepatocyte-specific *ABCA1* knockout mice, which demonstrated an increase in hepatic LDLR in the absence of ABCA1 that led to increased catabolism of plasma HDL particles [73]. In contrast, another study showed that ABCA1 overexpression in the liver of LDLR knockout mice led to the accumulation of atherogenic lipoproteins and atherosclerosis that was associated with rapid transfer of free cholesterol from HDL particles to apoB-lipoproteins [74]. We also found that the LXR agonist T0901317 inhibited the entry of SARS-CoV-2pp into cells. This finding is in agreement with previous studies demonstrating that pharmacological stimulation of ABCA1 inhibits HCV infection by affecting virus entry into cells [75]. Similarly, HIV replication has been shown to be impaired when ABCA1 was upregulated by LXR agonists [76,77], and this effect was mainly attributed to a reduced number of lipid rafts and depletion of membrane cholesterol, resulting in a reduction in the infectivity level of HIV.

The results of this study, while surprising and interesting, show, for the first time, that the use of several independent approaches to modulate cellular cholesterol revealed a differential effect between the use of cholesterol-sequestering agents and cholesterol-lowering drugs on SARS-CoV-2pp entry into cells (Figure 10). Depletion of cholesterol-rich rafts by either MβCD or nystatin enhanced the entry of both pseudotyped and wild-type viruses, most likely due to disruption of plasma membrane integrity associated with the possible dysregulation, loss, or shift of receptors from raft to non-raft domains. Consequently, SARS-CoV-2 is no longer in need of cholesterol-rich rafts to interact with host cells, allowing an elevated amount of the virus to freely enter the host through altered membrane rafts. On the other hand, when intracellular cholesterol is reduced by statins and siRNA-silencing cholesterol genes, SARS-CoV-2 entry into cells is significantly reduced due to the less abundant rafts and unavailability of cholesterol to facilitate the fusion of the spike protein with its receptor.

Collectively, our data are in support of the use of cholesterol-lowering medications by COVID-19 patients; however, focused randomized controlled trials are needed to prove statins’ beneficial effect in reducing SARS-CoV-2 infection. Although the molecular mechanisms need to be studied further, our study provides clear evidence of the implication of SREBP pathway activation in regulating host cholesterol hemostasis to promote lipid raft formation for productive virus entry. Such a mechanism of SARS-CoV-2-induced activation of SREBP2 may occur through oxidative stress-associated pathways [78,79]. Nevertheless, future investigations should focus on how SARS-CoV-2 could alter cellular cholesterol metabolism to hijack cholesterol, modulate various intracellular fluxes (in/out), and specifically exploit cholesterol to enter target cells. Promising avenues to pursue also include the possibility of a contribution to this process by other components of rafts including sphingolipids [80], which might shed light on the exact mechanism(s) underlying cholesterol-rich rafts mediating SARS-Cov-2 entry. This may lead to a better understanding of virus–host interactions in the initial steps of viral infection.

In conclusion, the present study clearly documents a central role of cellular cholesterol in the first step of SARS-CoV-2 infection. In particular, our work provides new insights underpinning the requirement of cholesterol-rich membrane raft integrity and intracellular cholesterol homeostasis for SARS-CoV-2 entry in host cells. Characterization of the molecular pathogenesis of SARS-CoV-2 may provide new additional therapeutic strategies to prevent SARS-CoV-2 infection.

## Figures and Tables

**Figure 1 nutrients-14-03417-f001:**
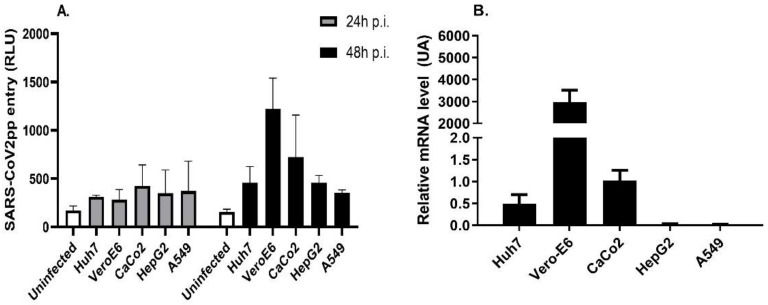
The efficiency of SARS-CoV-2pp entry into cells depends on the level of cellular *ACE* expression. Different cell lines were infected with SARS-CoV-2pp in complete growth media for 24 h and 48 h. Luciferase activity (expressed in relative light units, RLUs) was measured at 24 h p.i. and 48 h p.i. (**A)**. In a separate experiment, cells were used at 48 h p.i. for total RNA extraction to measure mRNA expression levels of the ACE2 receptor by qRT-PCR (**B**). Values are plotted as the mean ± SD.

**Figure 2 nutrients-14-03417-f002:**
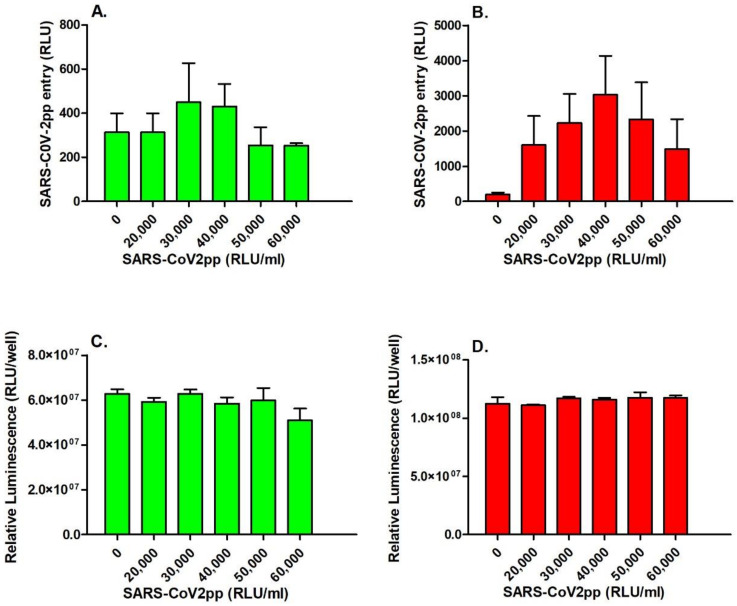
SARS-CoV-2pp enter the cells in a dose-dependent manner without altering cell viability. Vero E6 and Huh7 cells were infected with increasing doses of SARS-CoV-2pp (RLU/mL). At 48 h p.i., SARS-CoV-2pp entry into cells was assessed by a luciferase activity assay (RLUs) in Hu7 cells (**A**) and Vero E6 cells (**B**), and cell viability was measured by a luminescence assay for quantification of ATP levels in metabolically active Hu7 (**C**) and Vero E6 cells (**D**). Data for cell viability are expressed as relative luminescence units (RLUs). Values are plotted as the mean ± SD.

**Figure 3 nutrients-14-03417-f003:**
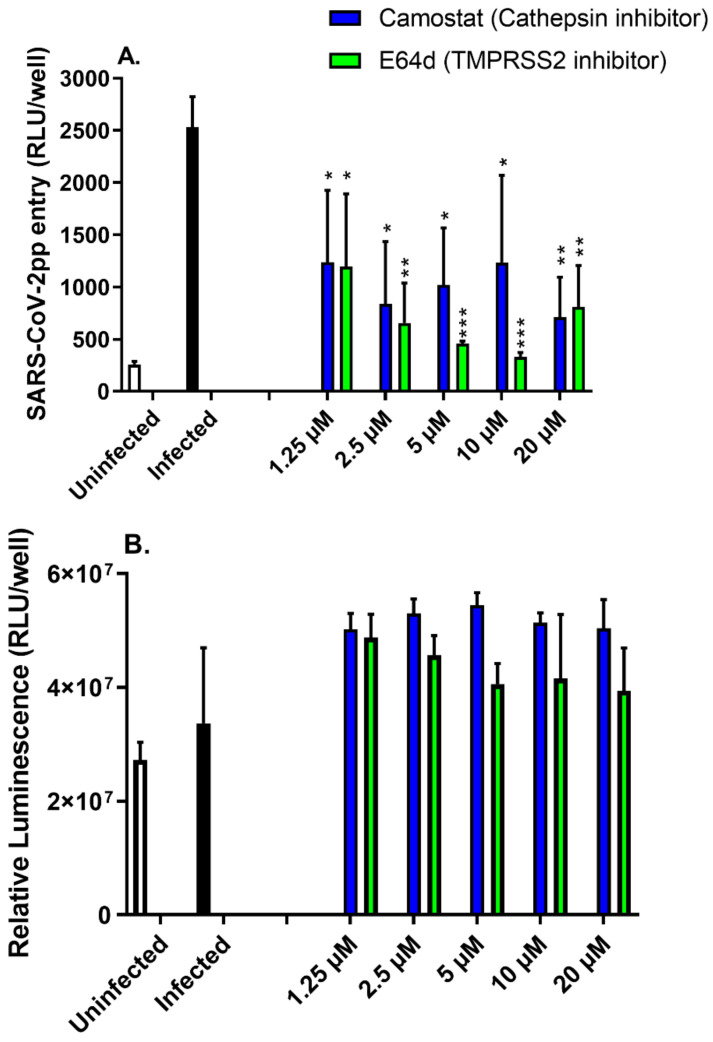
Protease inhibitors inhibit SARS-CoV-2pp entry in Huh7 cells. Huh7 cells were treated with increasing doses of either camostat mesylate or E-64d for 24 h prior to infection with SARS-CoV-2pp (30,000 RLU/mL). At 48 h p.i., cells were used for a luciferase activity assay (**A**) and a cell viability assay (**B**). Values are plotted as the mean ± SD (N = 4). *p* values were calculated using a two-tailed Student *t*-test. *, *p* < 0.05; **, *p* < 0.001; and ***, *p* < 0.0001.

**Figure 4 nutrients-14-03417-f004:**
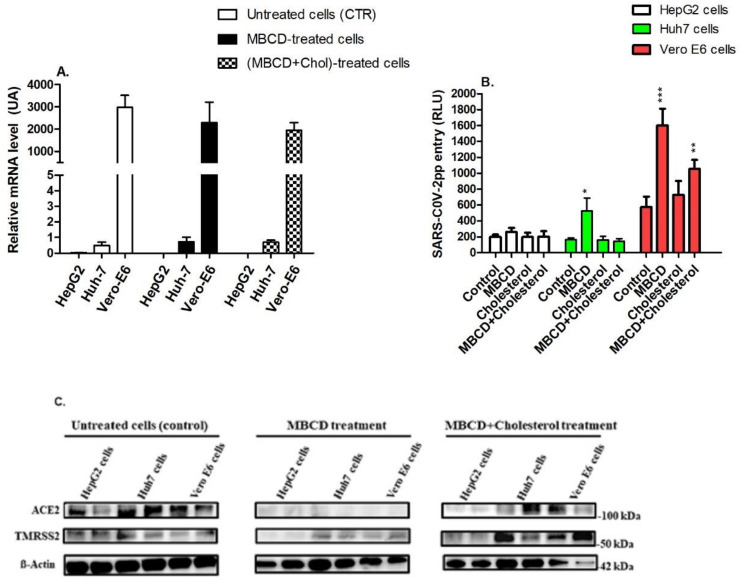
Depletion of cellular cholesterol alters raft-associated receptors involved in SARS-CoV-2pp entry. Different cells lines were treated with MβCD or MβCD-cholesterol complexes for 1h prior to SARS-CoV-2pp infection. At 48 h p.i., cells were used for the total RNA extraction to measure the expression of the ACE receptor by qRT-PCR (**A**). In separate experiments, cells were used to measure SARS-CoV-2pp entry using a luciferase assay (**B**) and to determine the protein expression levels of the ACE2 receptor and TMPRSS by Western blotting (**C**). Values are plotted as the mean ± SD (N = 3). *p* values were calculated using a two-tailed Student *t*-test. * *p* < 0.05, ** *p* < 0.001, and *** *p* < 0.0001 vs. control group.

**Figure 5 nutrients-14-03417-f005:**
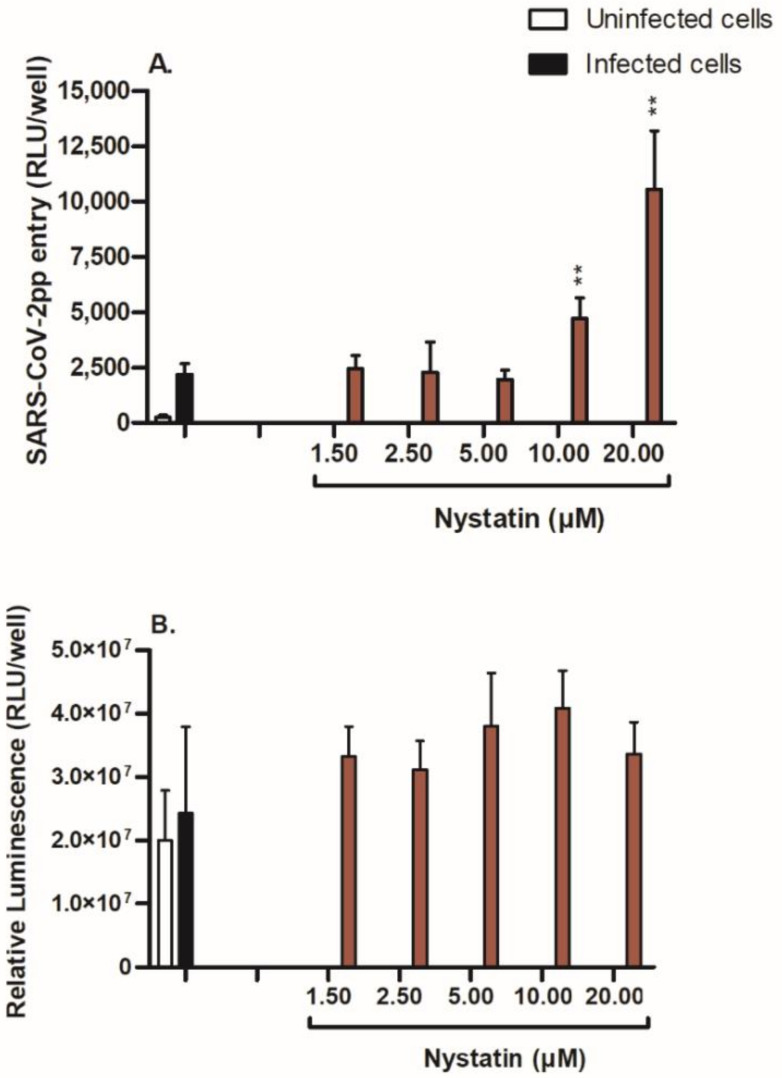
Effect of nystatin on SARS-CoV-2pp entry into Huh7 cells. Huh7 cells were treated with increasing concentrations of nystatin (1.5–20 µM) prior to SARS-CoV-2pp infection. At 48 h p.i., cells were used for a luciferase assay to measure SARS-CoV-2pp entry into cells (**A**), and for a cell viability assay as assessed by cellular ATP levels (**B**). Values are plotted as the mean ± SD (N = 4). *p* values were calculated using a two-tailed Student *t*-test. * *p* < 0.05 and ** *p* < 0.001 vs. infected group (control).

**Figure 6 nutrients-14-03417-f006:**
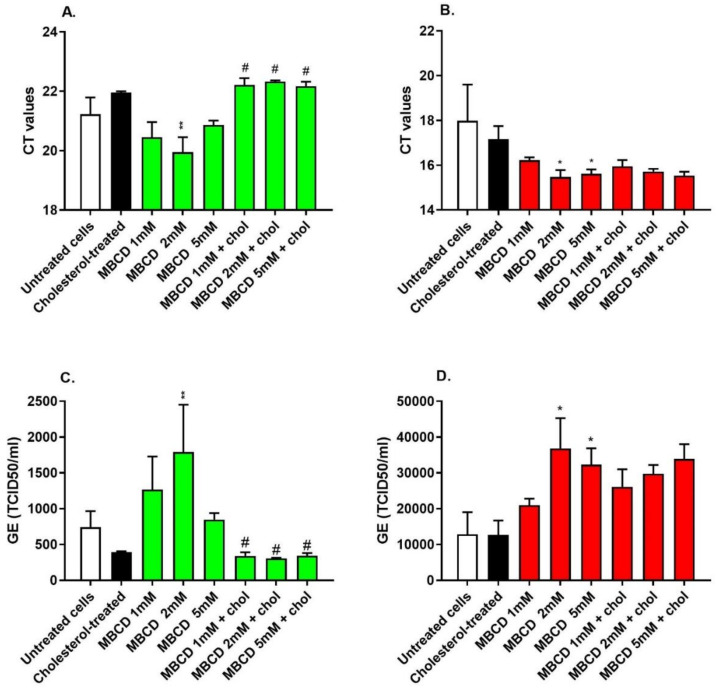
MβCD treatment enhanced wild-type SARS-CoV-2 infectivity in Huh7 and Vero E6 cells. Huh7 and Vero E6 cell lines were treated with different concentrations of MβCD (1 mM, 2 mM, and 5 mM) or cholesterol (0.25 mM) for 1 h. At 48 h p.i., Huh7 cells (**A**,**C**) and Vero E6 cells (**B**,**D**) were used for the extraction of total RNA, and cDNA was probed with ORF1a primers using a qRT-PCR assay to determine Ct values (**A**,**B**) and the cellular viral load (**C**,**D**). SARS-CoV-2 GE (TCID50/mL) represents the genome equivalent (GE) to the titer of 50% tissue culture infectious dose per ml. Values are plotted as the mean ± SD (N = 3). *p* values were calculated using a two-tailed Student *t*-test. * *p* < 0.05 and ** *p* < 0.001 vs. untreated control group; # *p* < 0.05 vs. MBCD-treated group.

**Figure 7 nutrients-14-03417-f007:**
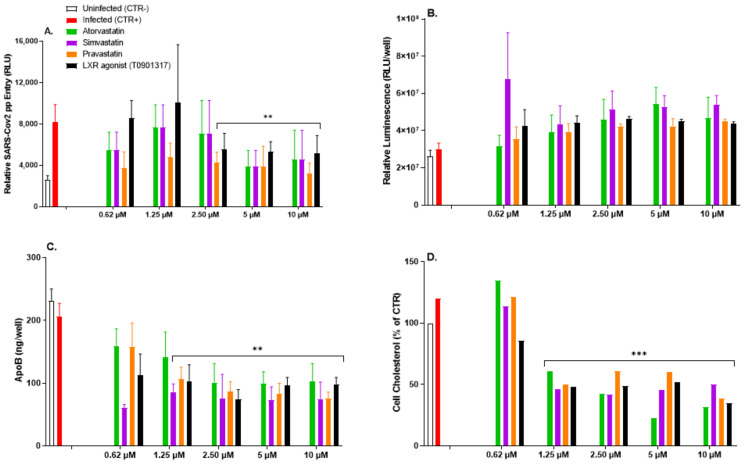
Alteration in intracellular cholesterol by statins reduces *SARS-CoV-2pp* entry into cells. Huh7 cells were treated with increasing concentrations of various classes of statins and the LXR agonist T0901317 for 48 h prior to cell infection with SARS-CoV-2pp (35,000 RLU/mL). At 48 h p.i., cells were used to measure the virus entry into cells by a luciferase assay (**A**), cell viability (**B**), apoB-LDL by ELISA (**C**), and the cellular cholesterol content (**D**). Values are plotted as the mean ± SD (N = 3, except for cholesterol content, N = 2). *p* values were calculated using a two-tailed Student *t*-test. ** *p* < 0.001, and *** *p* < 0.0001 vs. infected group (CRT+).

**Figure 8 nutrients-14-03417-f008:**
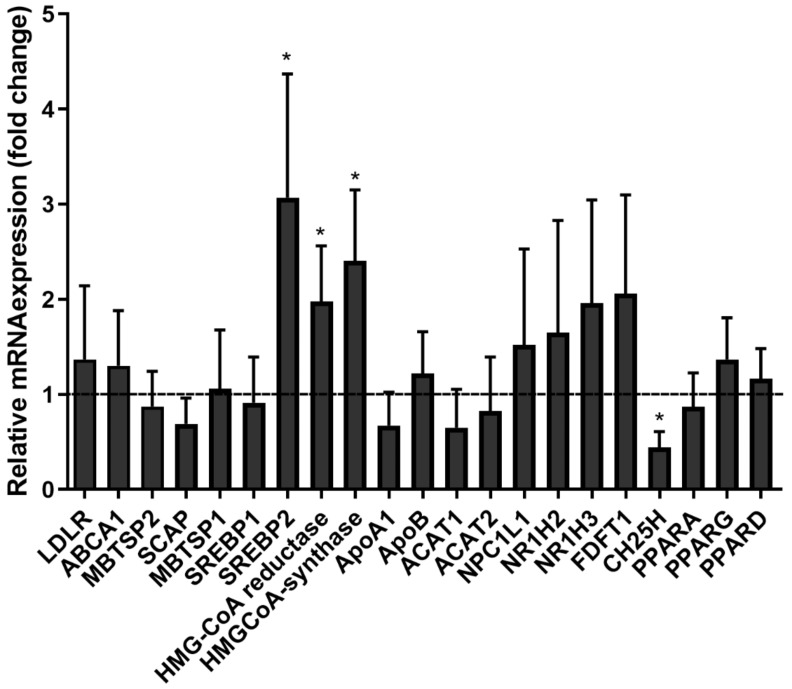
SARS-CoV-2pp entry into cells modulates the expression of genes involved in cholesterol biosynthesis. Huh7 cells were infected with SARS-CoV-2pp (35,000 RLU/mL) for 48 h. Infected cells and uninfected cells (control) were used for extraction of total RNA to measure the expression of various genes by qRT-PCR. Values are plotted as the mean ± SD of three independent experiments. *p* values were calculated using a two-tailed Student *t*-test. *, *p* < 0.05. Abbreviations: LDLR, low-density lipoprotein; ABCA1, ATP binding cassette A1; SCAP, SREBP cleavage-activating protein; MBTSP2, membrane-bound transcription factor peptidase site 2; SREBP1, sterol regulatory element binding transcription factor 1; SREBP2, sterol regulatory element binding transcription factor 2; HMG-CoA reductase, 3-hydroxy-3-methyl-glutaryl-coenzyme A reductase; HMG-CoA synthase, 3-hydroxy-3-methyl-glutaryl-coenzyme A reductase; ApoA1, apolipoprotein A1; ApoB, apolipoprotein B; ACAT1, acetyl-CoA acetyltransferase 1; ACAT2, acetyl-CoA acetyltransferase 2; NPC1L1, Niemann-Pick C1-like 1; NR1H2, nuclear receptor subfamily 1group H member 2; NR1H3, nuclear receptor subfamily 1group H member 3; FDFT1, farnesyl diphosphate farnesyltransferase 1; CH25H, cholesterol-25 hydroxylase; PPARA, peroxisome proliferator-activated receptor alpha; PPARG, peroxisome proliferator-activated receptor gamma; PPARD, peroxisome proliferator-activated receptor delta.

**Figure 9 nutrients-14-03417-f009:**
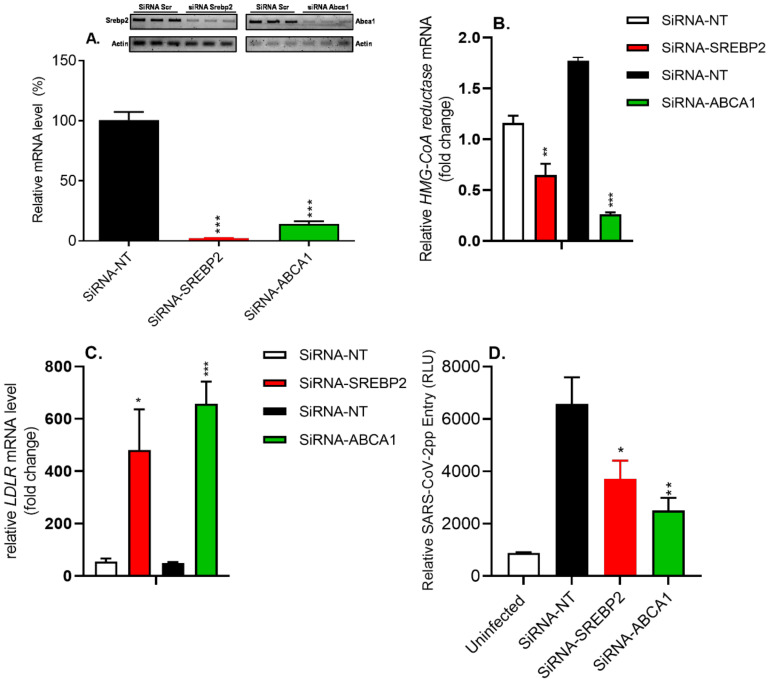
Silencing of the *SREBP2* and *ABCA1* genes reduces *SARS-CoV-2pp* entry into cells. Huh7 cells were transfected with siRNA-NT (control) or siRNAs targeting *SREBP2* and *ABCA1* for 48 h prior to infection with SARS-CoV-2pp for an additional 48 h. The relative quantity of mRNAs for *SREBP2* and *ABCA1* was determined by qRT-PCR (**A**), and the qPCR products were separated on a 2% agarose gel and visualized (Panel **A**, top). Additionally, targets genes for the knockdown of *SREBP2* and *ABCA1* were quantified by qRT-PCR (**B**,**C**). SARS-CoV-2pp entry was assessed by a luciferase assay (**D**). Values are plotted as the mean ± SD (N = 4). *p* values were calculated using a two-tailed Student *t* test. * *p* < 0.05, ** *p* < 0.001, and *** *p* < 0.0001 vs. siRNA-NT group (control).

**Figure 10 nutrients-14-03417-f010:**
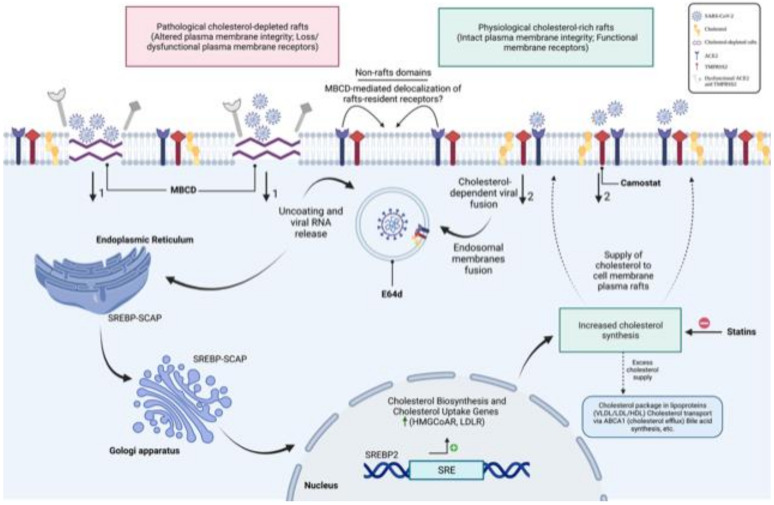
Hypothetical schematic representation depicting the importance of cellular cholesterol homeostasis in SARS-CoV-2 entry in host cells. Entry of SARS-CoV-2 occurs through the binding of the viral spike protein with the host ACE2 receptors facilitated by TMPRSS2 and the possible interaction with the cholesterol of raft domains. Once the viral RNA is released, the viral DNA is formed by integration with the host DNA and replication, leading to synthesis of mRNA and production of structural and non-structural proteins in the ER/Golgi compartments. Cholesterol-depleting agents such as MβCD or nystatin (relatively high doses) result in disruption of cell membrane integrity associated with membrane receptor dysregulation, leakage, or shift of ACE2 receptors to non-raft domains, leading to enhanced entry of SARS-CoV-2 particles into the cells (*arrow 1*). The definition “pathological rafts” adopted in this schematic is not related to any of the known pathological diseases; it is used for the purpose to emphasize the degree of structural disruption of lipid rafts caused by cholesterol-rich raft-sequestering agents. In contrast, when sufficient cholesterol is available in the raft domains, SARS-CoV-2 binds to the ACE2 receptor and efficiently enters the host (*arrow 2*) for productive infection. For simplicity, other residents of the plasma membrane such as sphingolipids are not illustrated in this diagram. Possible synergy between cholesterol and sphingolipids may influence virus entry in the host due to the high affinity of sphingolipids for cholesterol in the membrane rafts. SARS-CoV-2 may hijack cellular cholesterol to cause a reduction in cholesterol levels in the ER and consequently initiate the SREBP/SCAP complex to move from the ER to the Golgi where two proteases cleave SREBP. After cleavage, the NH2-terminal region of SREBP translocates to the nucleus to activate the transcription of target genes involved in cholesterol biosynthesis and transport. The large amount of newly synthesized cholesterol serves as a supply for the plasma membrane, and the excess of cholesterol is packaged into lipoproteins or effluxes by the cholesterol transporter ABCA1 or used for bile synthesis. How SARS-CoV-2 could hijack intracellular cholesterol to potentially modulate cholesterol fluxes (in/out of the host) needs to be explored.

## Data Availability

Not applicable.

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
