# Peer review of "Lipid Raft Integrity and Cellular Cholesterol Homeostasis Are Critical for SARS-CoV-2 Entry into Cells"

_nutrients, 2022, doi:10.3390/nu14163417_

Round 1

Reviewer 1 Report

The authors Bakillah et al. have submitted a manuscript titled "Lipid raft integrity and cellular cholesterol homeostasis are critical for SARS-CoV-2 entry into cells" as an original research article. The manuscript explores the role of intracellular cholesterol levels in regulating the entry of SARS-CoV-2 into the cells.

The authors have tested this hypothesis using various concentrations of SARS-CoV-2 and in different cell types before narrowing down on Huh7 and Vero-E6 cells for detailed studies. Next, the authors analyzed the effect of agents known to suppress intracellular cholesterol from functioning by preventing its synthesis or affecting its efflux from the cells. The authors have provided figures to support the description of the results. However, the Authors could incorporate several points to improve the scientific quality of the manuscript.

Major comments

·         In the abstract and figure 4, the authors mention and demonstrate that decrease in levels of ACE2 expression results in higher levels of SARS-CoV-2 viral entry into the cells. Authors need to explain these results in the context of the multitude of published literature that has shown that levels of ACE2 and SARS-CoV-2 infection are proportional to each other.

·         Section 3.1 - The title is opposite to the description as the authors state that VeroE6 has the highest infection due to its high ACE2 expression levels. Also, the expression levels of ACE2 in Huh7 are significantly lower than moderate. The authors should address this discrepancy in their interpretation of the results.

·         The authors should support their claim of ACE2 mRNA expression in all figures with protein level analysis similar to Figure 4C.

·         Interpretation of Figures 2A and 2B is incorrect. The levels of entry in Huh7 cells (Figure 2B) are ten times higher than in Vero E6 cells (Figure 2C).

·         In section 3.3, The authors need to cite their data or previous studies to show that MBCD treatment results in cholesterol depletion from the plasma membrane.

·         The levels of control in Figure 4B are significantly different compared to Figure 2A-B. The authors need to justify this significant difference.

·         Figure 4C does not include the ACE2 protein level expression data during only cholesterol treatment. The data on MBCD + Cholesterol treatment does not provide sufficient data to interpret this. Also, the levels of ACE2 are significantly different between the two replicates of the same cell and treatment type. It would be helpful to include a third replicate of the assay and do statistics on the western blot to confirm that the difference within the same treatment group is insignificant. Further, given the significant difference in actin levels across all the three treatment types, it would be helpful to use another housekeeping protein or total protein as a loading control.

·         The Y-axis for Figure 5B should reflect ATP expression. Also, the figure legend for Figure 5 needs to describe each treatment and readouts of each figure separately.

·         Figures 6A and 6B show that the viral load levels in MBCD-treated cells are significantly less than in untreated cells. Also, the authors should explain the difference between the amount of virus entering the cells vs. the viral load from the same cells. It would have been helpful had the authors used the same concentration of MBCD treatment for both viral entry and viral load studies. Further, the authors need to provide a more extensive description of Figures 6C and 6D as the current figure legend does not describe, for example - the meaning of the Y-axis title - GE (TCID50/ml).

·         In Figure 7, The difference in fluorescence intensity of SARS-CoV-2 spike protein is not apparent in the representative images. It would be helpful if the authors could provide quantitative proof of the difference in fluorescence intensity or results of the three assays where they observed a significant difference in the fluorescence of the Spike protein.

·         In section 3.6, The authors need to establish a link about how statins affect intracellular cholesterol levels for the reader to better appreciate their line of questioning.

·         In figure 8A, the authors need to correct the interpretation of the decrease in viral entry levels upon 10uM statin treatment. In figure 8A, treating infected cells with LXR agonists reduced the levels of viral entry. This result goes against the authors' hypothesis in the previous figures that maintenance of intracellular cholesterol levels results in inhibiting viral entry. If LXR agonist treatment results in the efflux of cholesterol, there would be a decrease in intracellular cholesterol levels, affecting raft formation and enhancing viral entry into the cells. Also, The authors need to do appropriate statistics and depict them accurately to appreciate better the difference they are trying to demonstrate.

·         In figure 9, the authors state that SARS-CoV-2 infection results in upregulation of the genes involved in the cholesterol metabolic pathway. However, out of the 21 genes tested, only three increased significantly, and one decreased significantly. Therefore, the authors need to provide a more nuanced interpretation of this result by focusing on the genes on a specific pathway like cholesterol bio-synthesis or efflux pathway.

·         In Figure 10, the authors selected to knock down SREBP2 and ABCA1. The authors need to justify their decision to knock down these two genes as SREBP1 and SREBP2 are known to be redundant, and ABCA1 was not significantly up or downregulated in Figure 9. Also, since SBREBP2 is a transcription factor, the authors should demonstrate changes in the active form of the protein and the results changes in the functional forms of its downstream proteins like HMG-CoA reductase and LDLR.

Minor comments:

·         Authors could list the primers as a table.

·         The viral nomenclature should be uniform throughout the manuscript, including figures and figure legends. If used, authors should describe the variant strains of the virus accordingly. Also, the authors need to explain the difference between the pseudotype virus and the actual virus.

·         In section 3.1, the authors must clarify if they mean 1 X 10^7 or 5 X 10^7 concerning virus yield.

·         Authors need to provide a citation for introductory statements of each result sub-section.

Author Response

See attached document. Thanks

Reviewer 2 Report

Dear authors,

This original manuscript focuses on the importance of lipid raft integrity and cellular cholesterol homeostasis for SARS-CoV-2 entry into cells. This is a very interesting research, and the article is well written and structured.

Revision has been performed on the paper's importance, originality and clarity, the study's validity and its relevance to the remit of the journal. 

Best regards,

Author Response

See attached document. Thanks
